# Optimization of Copper Stain Removal from Marble through the Formation of Cu(II) Complexes in Agar Gels

**DOI:** 10.3390/gels7030111

**Published:** 2021-08-06

**Authors:** Antonio Sansonetti, Moira Bertasa, Cristina Corti, Laura Rampazzi, Damiano Monticelli, Dominique Scalarone, Adele Sassella, Carmen Canevali

**Affiliations:** 1Institute for Heritage Science (ISPC), National Research Council (CNR), 20122 Milan, Italy; antonio.sansonetti@cnr.it (A.S.); laura.rampazzi@uninsubria.it (L.R.); 2Department of Chemistry and NIS Centre, University of Turin, 10125 Turin, Italy; MBertasa@britishmuseum.org (M.B.); dominique.scalarone@unito.it (D.S.); 3Department of Human Sciences and Innovation for the Territory, Università degli Studi dell’Insubria, 22100 Como, Italy; cristina.corti@uninsubria.it; 4Department of Science and High Technology, Università degli Studi dell’Insubria, 22100 Como, Italy; damiano.monticelli@uninsubria.it; 5Department of Materials Science, University of Milano-Bicocca, 20125 Milan, Italy; adele.sassella@unimib.it

**Keywords:** agar gel, EDTA, TAC, alanine, UV-Vis spectroscopy, EPR

## Abstract

Copper complexes with different ligands (ethylenediaminetetraacetic acid, EDTA, ammonium citrate tribasic, TAC, and alanine, ALA) were studied in aqueous solutions and hydrogels with the aim of setting the optimal conditions for copper stain removal from marble by agar gels, with damage minimization. The stoichiometry and stability of copper complexes were monitored by ultraviolet-visible (UV-Vis) spectroscopy and the symmetry of Cu(II) centers in the different gel formulations was studied by electron paramagnetic resonance (EPR) spectroscopy. Cleaning effectiveness in optimized conditions was verified on marble laboratory specimens through color variations and by determining copper on gels by inductively coupled plasma-mass spectrometry (ICP-MS). Two copper complexes with TAC were identified, one having the known stoichiometry 1:1, and the other 1:2, Cu(TAC)_2_, never observed before. The stability of all the complexes at different pH was observed to increase with pH. At pH 10.0, the gel’s effectiveness in removing copper salts from marble was the highest in the presence of ALA, followed by EDTA, TAC, and pure agar gel. Limited damage to the marble surface was observed when gels with added EDTA and TAC were employed, whereas agar gel with ALA was determined to be the most efficient and safe cleaning material.

## 1. Introduction

Cleaning procedures are considered the most delicate and irreversible phase in a restoration work and they should fulfil the fundamental requirements of high effectiveness and restrained damage. Several cleaning issues are currently open and metal stain removal is one of the unsolved ones [1,2,3,4]. Metal stains occur when bronze or iron components, such as sculptures, clamps, pivots, or plaques are in contact with stone ashlars releasing corrosion products, usually showing a greenish or reddish color, respectively. The issue is particularly serious when the stains hinder the correct readability of an artwork [5,6,7], as in the case of light-colored and/or porous stone materials (limestones, sandstones, calcarenites, travertines, and marbles) used in sculptural artworks (statues, busts, friezes, basements, etc.) and in buildings. Corrosion processes are in turn strictly dependent on the chemical composition, structure, and metallurgical characteristics of the metal artifacts and on surrounding environment [8]. The current practices to clean marbles (sandblasting, laser, nebulized water spray) do not show satisfactory effects on metal stains. In these cases, conservators should attempt a chemical extraction, for instance by using hydrogels.

Hydrogels contain hydrophilic groups capable of adsorbing large quantities of metal ions and are already used for wastewater treatment [9,10,11].

Chelating agents have already been successfully used in aqueous solution [8] and in hydrogels (from now on termed gels) [5,12]. Chelating agents are also used for the removal of newly formed calcium compounds, such as black crusts mainly composed of gypsum and calcite, and white calcite thin layers or encrustations [13].

Understanding and increasing the effectiveness of these materials calls for the study of metal complex formation in water-based extraction systems and in gels, matrices that may induce different behavior in the complex formation. As an example, the level of metal cation detection and the time required for forming metal complexes are at least one order of magnitude more favorable in silica gel than in solution [14]. Moreover, the formation of metal complexes in agar gel allows for the effective removal of water soluble copper salts from marble surfaces [5,15,16]. These examples highlight the interest for the comparison between gels and free aqueous solutions, even if silica gel and Agar gel are very different, being the first a chemical gel and the second a physical gel [17]. Such a difference involves different gelation mechanisms and different properties, particularly concerning thermo-reversibility.

The cleaning method, which uses agar gel, is very versatile [15,18], low cost, and allows for a controlled water release [2,19,20,21,22], together with a low impact on the artwork. Many types of agar powders are commercially available [23] and agar is currently the most used material by Italian restorers facing specific cleaning problems on delicate artwork surfaces [1,2,3,4,16,24,25,26,27]. Agar gels are used pure or with ethylenediaminetetraacetic acid (EDTA), ammonium citrate tribasic (TAC), or alanine (ALA). Gels with EDTA and TAC displayed the best cleaning effectiveness for copper stain removal from marble surfaces, thanks to metal complexation [5]. ALA showed a high selectivity for copper ion removal in aqueous solution [8], but it has not yet been tested in gels.

It is known that EDTA, TAC, and ALA form stable metal complexes in free aqueous solutions. EDTA has six potential sites available for binding with metal cations, composed of four carboxyl and two amino groups [14], and forms a very stable 1:1 complex, with log K_f_ 18.78 for Cu(EDTA) [28]. Considering TAC, it is known that metal complexes of citric acid belong to the group of highly stable 6-membered ring complexes, i.e., log K_f_ = 5.03 for Cu_2_TAC_2_(OH)_2_ [29]. As for ALA complexes, their stability is due to the chelating ability of both carboxyl and amino groups and to the effectiveness of intramolecular interactions within the complexes [30], i.e., log K_f_ = 6.65 for Cu(ALA)_2_ [31].

It is known that metal complexes in aqueous solutions increase their stability with pH [28]. However, a study of metal complex stability within gels remains to be performed.

Notwithstanding its importance in restoration practice, the optimization of copper stain removal from marble by added gels has not yet been systematically considered. In the present paper, we systematically evaluated the stoichiometry and the stability of copper complexes with EDTA, TAC, and ALA, both in free aqueous solutions and in gels, with special emphasis on the effect of pH. Here, the stoichiometric ratios of copper complexes was verified by the mole ratio method at maximum wavelength [32] and the stability of copper complexes was studied by ultraviolet-visible (UV-Vis) spectroscopy and electron paramagnetic resonance (EPR) spectroscopy. The cleaning effectiveness of gels in optimized conditions was assessed by quantifying the amount of removed copper by inductively coupled plasma-mass spectrometry (ICP-MS) and by the color variations of marble laboratory specimens homogeneously stained with brochantite. Possible damage to the marble surface was evaluated by the amount of calcium removed by the gels.

## 2. Results and Discussion

### 2.1. Characterization of Copper Complexes in Aqueous Solutions and in Gels

#### 2.1.1. UV-Vis Characterization

The stability of copper complexes prepared at the proper metal:ligand (EDTA, TAC and ALA) ratio was investigated at different pH values, firstly in aqueous solutions, then in gels. With this aim, the stoichiometric ratio of the Cu(II)-ligand complexes was first verified using the mole ratio method at maximum wavelength [32]. Complex stability was studied by monitoring the absorption of their metal-ligand charge transfer (MLCT) strong bands in the 190–400 nm range. The spectral positions λ_max_ for all solutions and ligands are summarized in Table 1, together with the attributions to specific chemical species. Finally, in all spectra, bands attributed to the d-d transition of Cu^2+^ were also detected at 800 nm, too weak and broad to be used for diagnostic purposes.

At first, spectra of pure agar gel and of CuSO_4_ in free solution and in agar gel were recorded to be used as reference, and are shown in Figure 1. The absorbance spectrum of pure agar gel shows no bands, but it consists of a long monotonous rise towards low wavelengths, probably due to light scattering by the agar gel itself. Indeed, we verified that it nicely follows a λ^−4^ law (see Appendix A). For CuSO_4_ in gel, a shoulder at about 233 nm is observed, becoming more evident by subtracting the spectrum of pure agar gel; thus, this band at 233 nm (dotted curve) can be attributed to copper centers coordinated by pure agar gel, which shall henceforth be termed Cu(agar), in agreement with the literature [5]. Finally, the spectrum of CuSO_4_ in free aqueous solution is also reported for comparison (dashed curve), displaying a broad band centered at about 198 nm, attributed to copper aquoion [Cu(H_2_O)_6_]^2+^ [33].

The assessment of the Cu(EDTA) 1:1 complex stoichiometry was accomplished by the spectrophotometric method [14]. It must be pointed out that the notation from now on used to indicate copper complexes only shows the stoichiometric ratio between the metal and the ligand, this last being indicated by the total concentration of the free ligand in all its forms (i.e., EDTA = Y^4−^ + HY^3−^ + H_2_Y^2−^ +….), in agreement with the literature [25]. This notation does not specify either the ligand protonation according to the ligand stability as a function of pH, or the electric charge of the complex. For this purpose, aqueous solutions containing CuSO_4_ and EDTA at increasing molar ratios (from 1:1 to 1:10) were prepared and the UV-Vis spectra are reported in Figure 2a. The 1:1 solution displayed a band at 239 nm [34], which remained unchanged by increasing the ligand content up to 1:10 molar ratio (see the arrow in Figure 2a). Zoom lines confirm the negligible difference among the different curves. The band at 239 nm is well separated from those of the precursors in free aqueous solution. Indeed, EDTA shows a maximum at 191 nm (dotted curve), which was experimentally observed to increase in intensity (see Appendix A), with a broadening and shift towards higher wavelengths as EDTA concentration increases, while CuSO_4_ displays the MLCT band at 198 nm (dashed curve). In free aqueous solution, a Cu(EDTA) = 1:1 complex is therefore demonstrated to form, as indeed reported in the literature [14,28], and it originates in the 239 nm absorption band.

In order to study complex stability with pH in a free aqueous solution, the absorbance spectra of Cu(EDTA) 1:1 were recorded, both ‘as prepared’ (pH 5.0), and after the addition of ammonium carbonate in different amounts until pH 6.0 and 8.0 were reached (see Appendix A). The spectra confirm that a Cu(EDTA) complex forms in acidic medium, in agreement with the literature [35]. Furthermore, the intensity at λ_max_ =239 nm gradually increases with pH; this matched well with the increase in EDTA complex stability with pH. The absorbance at 191 nm, attributed to the pure ligand, also increases with pH, as observed in solutions of pure EDTA (spectra not shown), probably because of an increase in ligand molar absorptivity originating from the formation of differently protonated EDTA species [28]. The agar gel containing an aqueous solution of CuSO4 and EDTA (in molar ratio M:L = 1:1) shows a pH value of 6.0 and, after gel formation, a shoulder in the same position (λmax = 239 nm) as the band observed in the aqueous solution ([34], see Figure 2b). Such a shoulder increases in intensity at pH 10.0, suggesting an increased stability of the complex in the gel. This result was confirmed by subtracting the spectrum at the lower pH from the spectrum at the higher pH. For our gels it was not easy to detect the spectral shift observed in the literature for the MLCT transitions of metal-EDTA complexes in a sol-gel matrix with respect to solution [14]. Nevertheless, the shoulder can be attributed to the Cu(EDTA) complex, as it increases with the equimolar concentration of both CuSO4 and EDTA in agar gel.

In a solution with a 1:1 molar ratio of CuSO_4_ and TAC, the UV-Vis absorption spectrum dramatically changes with respect to the spectra of the precursor aqueous solutions and displays a structured broad band, centered at about 240 nm (Figure 3a). By gradually increasing the molar ratio, at first, with M:L = 1:2, two bands at 225 nm and at 260 nm resolved, suggesting the presence of two copper complexes with TAC. Then, further increasing molar ratio up to 1:5, the band at 225 nm was gradually hidden by the increasing ligand band, while the band at 260 nm remains constant. Zoom lines confirm the negligible difference among the different curves. This suggests that the band at 260 nm can be attributed to a second complex having a stoichiometry of 1:2, Cu(TAC)_2_, never observed before for Cu(II) complexes with TAC. A similar 1:2 stoichiometry was observed in the literature for TAC complexes with Pb(II) [36] and with In(III) [37]. Finally, the band at 225 nm could be due to the 1:1 complex, monomer Cu(TAC), or dimer Cu_2_(TAC)_2_ [29].

The aqueous solution containing CuSO_4_ and TAC at a molar ratio of 1:2 has pH 9.0. It was verified that the intensity at λ_max_ =260 nm for this solution decreases with pH from A_max_ = 0.8 at pH 9.0 to A_max_ = 0.5 at pH 6.0 (spectra not shown). A similar dependence of absorbance on pH is also observed in gel (Figure 3b). The agar gel containing an aqueous solution of CuSO_4_ and TAC (in molar ratio M:L = 1:2) has pH 5.5, and shows a shoulder at 260 nm, which increases in intensity by adding NH_3_ until pH 10.0; no other modifications were observed in the spectrum with pH. The shoulder at 260 nm is in the same position as the band observed in the corresponding aqueous solution in Figure 3a, suggesting the formation of the complex Cu(TAC)_2_ also in the gel. In addition, it cannot be excluded that a second band is present underneath the strong ligand absorption, corresponding to the 1:1 complex. Moreover, the comparison with the Cu(agar) complex spectrum (dashed curve) shows that this latter complex does not form in the presence of TAC.

The equimolar aqueous solution containing CuSO_4_ and ALA has pH 6.0 and its UV-Vis spectrum displays a broad band (spectrum not shown), which does not match with the response of the precursors aqueous solutions containing separately CuSO_4_ and ALA (dashed and dotted curves in Figure 4a, respectively). This band does not vary with the ligand concentration. A strong absorbance band at 234 nm appears at pH 10.0 for M:L 1:1 (Figure 4a), suggesting that such a high pH value favors the formation of the complex, since both the carboxylic and the amino groups of ALA are deprotonated and available for coordinating copper centers at quadrangular sites [38]. By increasing the ligand content from M:L 1:1 to 1:2 (see the arrow in Figure 4a) and maintaining pH at 10.0, the absorbance maximum increases. Then, from M:L = 1:2 it remains constant. Zoom lines confirm the negligible difference among the different curves. This result shows that a 1:2 copper-alanine complex, Cu(ALA)_2_, forms, in agreement with the literature [31].

The increase of absorbance with pH, already observed in free aqueous solution, is also visible in gels, as shown in Figure 4b. The agar gel containing an aqueous solution of CuSO_4_ and ALA (in molar ratio M:L = 1:2) has pH 3.4 and shows a small shoulder centered at about 234 nm, which strongly increases in intensity at pH 10.0. No other modifications were observed with pH. Even if the band almost saturates at pH 10.0 and the spectral position cannot be precisely read, it matches the position of the band observed in the aqueous solution containing CuSO_4_ and ALA in a molar ratio of 1:2 and attributed to the Cu(ALA)_2_ complex; this clearly supports our attribution. The UV-Vis spectrum of the Cu(agar) complex is also shown for comparison, as a dashed curve.

#### 2.1.2. EPR Characterization

Once verified by UV-Vis spectroscopy, the best conditions for Cu complex formation, the symmetry of Cu(II) centers within the different gel formulations was studied using EPR spectroscopy at 123 K. As expected, the large isotropic signal at g~2 of copper aqueous ions was observable in all aqueous solutions (spectra not shown) [36,37]. In pure gels, a weak sharp isotropic signal was present at g = 2.00 (indicated by the asterisk * in Figure 5), probably due to impurities present in the agar network, none of them attributable to the Cu(II) centers.

The EPR spectra of gels containing aqueous solutions of CuSO_4_ and the ligands (EDTA, ALA, and TAC) at the molar ratio requested by the stoichiometry of the complex (1:1 for EDTA; 1:2 for ALA and TAC) at pH 10.0 are reported in Figure 5. Their spectral parameters are reported in Table 2. Agar gel with CuSO_4_ shows a weak axial signal (Figure 5a) having values of g and A tensor components (g_1_~2.3; A_1_~140 G; g_2_ = g_3_~2.07) consistent with those of Cu(II) centers in a tetragonal symmetry field of oxygen atoms [39]. Accordingly, this signal can be attributed to magnetically diluted Cu(II) centers coordinated by agar gels, Cu(agar). For agar gel with CuSO_4_ and EDTA, a strong axial signal is observed, having g_1_ = 2.29; A_1_ = 141 G; g_2_ = g_3_ = 2.07 (Figure 5b). These spectral parameters are in agreement with those observed for the same gel at pH 5.0 [5] and attributed to the Cu(EDTA) complex. However, the Cu(agar) signal is no more observable at pH 10.0, meaning that the ligand outcompetes the agar binding sites at higher pH.

For agar gel with CuSO_4_ and ALA, an anisotropic signal is observed (g_1_ = 2.18, g_2_ = 2.07, g_3_ = 2.00, A_1_ = 183 G, A_2_ = 12 G), which shows a completely different coordination with respect to Cu(EDTA) complex (Figure 5c).

For agar gel with CuSO_4_ and TAC, two anisotropic signals were observed (Figure 5d), having the same spectral parameters, except g_2_ and A_2_ values. These two signals could be attributed to complexes having molar ratios Cu:TAC = 1:1 and 1:2, respectively, in agreement with the UV-Vis results. The comparison of the spectra obtained at pH 10.0 and at pH 7.0, this latter reported in the literature [5], suggests that copper centers are far better coordinated by TAC at higher pH values, since at pH 7.0 the signals of both the Cu(agar) complex and the interacting Cu(II) centers were observed. Finally, the presence of the dimer Cu_2_(TAC)_2_ cannot be confirmed by EPR spectra, since the spin coupling in the binuclear complex leads to the complete disappearance of the signal [29].

The complexes with ALA and with TAC have similar spectral parameters, displaying lower g_1_ and higher A_1_ values than Cu(EDTA). These variations can be related to a gradual increase in tetragonal distortion of copper centers in complexes with 1:2 stoichiometry [39,40].

### 2.2. Marble Specimens Cleaning

#### 2.2.1. ICP-MS Analysis of Gels

The amount of copper and calcium removed by gels from laboratory-stained surfaces was quantified by ICP-MS. Microscopic images of marble specimens before staining, after staining with brochantite, and after treatment with agar gel with EDTA, TAC and ALA for 60 min are reported in the Appendix A. All gels, as prepared, showed the presence of negligible copper and calcium amounts. The amount of copper removed from the surface area per gel unit at pH 10.0 is reported in Figure 6 as a function of gel composition (pure, added with EDTA, TAC or ALA) for two different contact times (30 and 60 min), beside the ‘as prepared’ gels (zero minutes). The removal of copper increases with contact time for all the investigated gels, but it also strongly depends on the composition of the gel. After 60 min of contact time, pure agar gel displays the lowest removal (10 μg/cm^2^), followed by gels with TAC (60 μg/cm^2^), EDTA (80 μg/cm^2^), and ALA (120 μg/cm^2^).

Interestingly, the copper concentration removed by pure agar gel is the same at pH 10.0 and at pH 5.0 [5], whereas the copper concentration removed by added gels is higher at pH 10.0. In fact, both the gels with EDTA and with TAC removed only 30 μg/cm^2^ at pH 5.0 and 7.0, respectively (Figure 6e), after 60 min contact [5]. These data show that the increase in pH did not affect removal by pure agar gel, while it did affects removal by the additives, in agreement with the UV-Vis results about the formation of complexes. Moreover, it is worth noting that the removal of copper by gels containing different additives at low pH values (pH 5.0 for EDTA and 7.0 for TAC [5]) showed no significant differences, probably due to the low stability of copper complexes.

Finally, the calcium content in gels, which may be removed due to the possible damage of the marble surface, was also analyzed. Pure agar gel and gel with added ALA removed no calcium from the specimens even after 60 min contact, while gels with EDTA and TAC removed 70 μg/cm^2^ of calcium after 30 min contact, with no further increase with time. These results confirm that copper removal is strongly influenced by the nature of the additive and that ALA shows a peculiar selectivity for copper in gels, as already observed in aqueous solution during travertine basement cleaning [8], with the 80% of copper extracted by both alanine and disodium EDTA, while calcium was only extracted by disodium EDTA. In particular, the best results were obtained using gels with ALA at pH 10.0, which allow a very good removal of copper salts from marble surface, with negligible calcium removal damage of the stone surface.

#### 2.2.2. Colorimetric Characterization of Marble Specimens

Color parameters of stained marble surfaces were measured before (t_1_) and after (t_2_) cleaning for different contact times (30 and 60 min) with gels at pH 10.0. The parameters extracted from the analysis are summarized in Table 3 and the calculated ΔE* values are also shown in Figure 7.

For pure gels, the smallest global color variations ΔE* were measured (ΔE* 2.61 and 5.25 after 30 and 60 min, respectively). These values are significant, since ΔE* > 2 corresponds to a cleaning action perceivable by naked eye [41]. After cleaning with the gels with additions, higher ΔE* values were measured, increasing in the order EDTA < TAC < ALA. This result confirms that added agar gels are more effective in removing copper compounds. In particular, the highest ΔE* value (ΔE* = 12.55) was achieved by the gel with ALA after 60 min contact, in agreement with the ICP-MS data on copper removal. Considering the differences in single color values (Table 3), it is possible to highlight that the highest contribution was given by Δa* (red/green variation), which reaches 10.57 after 60 min contact. This corresponds to a surface with less negative a* values, hence to the removal of green components. Finally, the action of the gels lightened the tested surfaces (ΔL* = 6.22 after 30 min contact with gel-ALA), even if this effect did not increase with application time.

## 3. Conclusions

The present paper describes the analysis of copper complex stoichiometry in free aqueous solutions and in gels with CuSO_4_ and different ligands (EDTA, TAC, ALA). The stability of the complexes was also studied, in view of their use for copper stain removal from marble and of process efficiency maximization with minimum damage. Complexes were prepared from the chelating agent and copper sulphate, the latter being chosen due to the need of having high concentrations of copper in water, which is not guaranteed if brochantite is used as a source of copper centers. The complexes were prepared at the proper metal: ligand ratio (1:1 for EDTA, 1:2 for TAC and ALA) and at different pH values (in all cases lower than neutrality and 10.0).

The stoichiometry of the Cu(EDTA) and of Cu(ALA)_2_ complexes was confirmed. Furthermore, two copper complexes with TAC were identified both in free aqueous solutions and in gels, one of them having the known stoichiometry 1:1, monomer Cu(TAC) or dimer Cu_2_(TAC)_2_, and the other 1: 2, Cu(TAC)_2_, which has never been observed before.

The stability of all the complexes with the ligands increased with pH, both in free aqueous solutions and in gels. Indeed, copper centers were better coordinated by gels and/or by additives at pH 10.0 than at lower pH values. Accordingly, the gel effectiveness in removing copper salts from marble is the highest in the presence of ALA, followed by EDTA, TAC and pure agar gel at pH 10.0. Color variations of marble laboratory specimens stained with brochantite and cleaned under well-controlled and reproducible conditions, fully confirmed these results. Damage of marble was observed only after cleaning with gels with added EDTA and TAC, so that agar gel with ALA at pH 10.0 can be deduced to be the most efficient agent for copper stain removal from marble, with minimum damage in terms of calcium removal. We believe that the obtained results are extremely useful for restorers interested in using more efficient gels for metal extraction.

## 4. Materials and Methods

### 4.1. Materials

AgarArt powder (CTS S.r.l., Milan, Italy) was used for gel preparation. Disodium EDTA (Merck-Millipore, Darmstadt, Germany), TAC (Bresciani S.r.l., Milan, Italy) and L-ALA (Sigma Aldrich, CAS n. 56-41-7, Darmstadt, Germany) were used as chelating agents. CuSO_4_·5H_2_O (Sigma Aldrich, Darmstadt, Germany) was used as a copper source. For laboratory specimen staining, Na_2_CO_3_ (Merck-Millipore, Darmstadt, Germany) was also used. For pH solution modification, (NH_4_)_2_CO_3_, H_2_SO_4_ 97% and NH_3_ 33% (Sigma Aldrich, Darmstadt, Germany) were used. HNO_3_ (70%, Carlo Erba, Cornaredo, Italy) was used for gel dissolution. MQ water^®^ and ultra-pure water were used when necessary.

### 4.2. Preparation of Solutions

Aqueous solutions at different metal:ligand molar ratios were prepared as folows: a solution of CuSO_4_·5H_2_O (5 × 10^⁻4^ M, i.e., 0.012% *w/w*) was added with increasing amounts of the ligand (EDTA, TAC, or ALA) to obtain the desired metal: ligand molar ratio.

In all cases, the concentration of solutions was chosen to observe intense, however not saturated, bands.

In order to compare all solutions at the same pH values, the pH values of the ‘as prepared’ solutions were varied in different ways (Table 1). For copper solutions containing EDTA and ALA, the initial pH values of 5.0 and 6.0, respectively, were increased to 6.0 and 8.0 for EDTA and to 10.0 for ALA by adding the proper amount of (NH_4_)_2_CO_3_. The copper solution containing TAC had pH = 9.0, which was decreased to 6.0 by adding the proper volume of H_2_SO_4_.

### 4.3. Preparation of Laboratory Specimens

Specimens sized 5 cm × 5 cm × 2 cm of white statuary Carrara marble were used. The most homogeneous surface (5 cm × 5 cm) of each specimen was selected and processed as previously described in the literature [5]. This surface was stained by in situ synthesis of brochantite, Cu_4_(SO_4_)(OH)_6_. For this purpose, marble specimens were immersed in 0.05 M (1.2% *w/w*) solutions of CuSO_4_·5H_2_O in water at 55°C, then an equimolar solution (i.e., 0.53% *w/w*) of Na_2_CO_3_ was added, dropwise. After the addition was complete, marble specimens remained immersed overnight into the reaction solution and were successively left to dry in the air. This procedure guaranteed reproducible staining in the light green palette, homogeneous on the whole specimen surface (Appendix A).

### 4.4. Preparation and Application of Agar Gels

Gels were prepared using the following method [7]: AgarArt powder (3% *w/w*) and CuSO_4_·5H_2_O (3 × 10^−3^ M, i.e., 0.075% *w/w*) were dissolved in water at boiling temperature, to form a colloidal solution. Additives (EDTA or TAC or ALA, each at 1%, *w/w*) were added to the aqueous solutions containing AgarArt powder and CuSO_4_·5H_2_O, when required, before heating. The colloidal solution gellifies when cooling at 35 °C, forming a rigid thermo-reversible gel that can be re-liquefied by heating it to about 80 °C. Thus, the formulation underwent a double heating, which ensured the formation of the double helix network in the gel, improving its homogeneity and transparency. At the end, the warm solutions (around 50 °C) were poured in Petri capsules (Ø = 115 mm) up to 0.5 cm thickness. When cooled, gels sized 2.5 × 2.5 × 0.5 cm were cut, applied to stained surfaces displayed horizontally, and left for 60 min. This procedure allowed treatment under well-controlled and reproducible conditions.

The concentration of the agar gel in solution was chosen according to the one giving good results in cleaning procedures according to the literature [1], where four different concentrations of agar in water were tested (1, 2, 3 and 4 *w/w*%). The additive concentration was chosen according to the conservators’ practical experience, which highlighted the improved cleaning action of the added gels at low ligand concentration [1]. Concentrations of metal and ligands in gels were chosen to observe intense, however not saturated, bands.

The pH values of the ‘as prepared’ solutions were 5.0 (pure agar), 6.0 (in the presence of EDTA), 5.5 (in the presence of TAC), and 3.4 (in the presence of ALA) (Table 1). To study the effect of pH on complexes in gel, solutions were added with the proper volume of NH_3_ 33% until reaching pH = 10.0. The maximum pH value was chosen at 10.0, since damage to the marble surface may occur above this value.

### 4.5. Solution and Agar Gel Characterization

The UV-Vis transmittance spectra of solutions and gels were recorded on an Agilent Cary 100 spectrophotometer in the spectral range from 190 to 900 nm, at 25 °C, in a thermostatic cell. Solutions were characterized in quartz cuvettes with 1 cm optical paths, then properly normalized to cuvette and pure solvent spectra. In the case of gels, UV-Vis transmittance spectra were recorded on gel layers ≈ 1 mm thick. The stoichiometric ratio of the Cu(II)-ligand complexes was assessed by using the mole ratio method at the maximum wavelength [32].

Electron paramagnetic resonance (EPR) spectra were recorded at the X-band frequency on a Bruker EMX EPR spectrometer equipped with a BVT 2000 (Bruker, Mannheim, Germany) variable temperature unit. Both solutions and gels were inserted into quartz tubes having an internal diameter of 3 mm. Since it was not possible to compact gels at the bottom of the EPR tubes, any quantitative comparison among different samples was avoided. Unless otherwise indicated, a modulation frequency of 100 kHz, modulation amplitude of 1 G, microwave power of 5 mW and temperature of 123 K were used. The g values were determined by standardization with α,α′-diphenyl-β-picryl hydrazyl (DPPH) radical.

In order to determine the amount of metal extracted from the specimens, gels (5 cm × 5 cm × 0.5 cm) were removed after 60 min contact, weighed, dissolved in HNO_3_, diluted, and analyzed using an inductively coupled plasma-mass spectrometer (ICP-MS, ThermoFisher iCAP Q). Ultra-pure water produced by a Sartorius arium mini system was used to dilute the samples and to prepare the standard solutions, and ultrapure nitric acid, produced via sub-boiling distillation with DuoPUR-Milestone equipment [42], was added to a final concentration of 2% *w/w*. Procedural blanks and control standards were also analyzed during each analysis batch: quantification was performed by external calibration. Data (± 2% error) were normalized according to each sample weight.

### 4.6. Marble Characterization

Color measurement data were acquired using a Konica Minolta Chromameter CM-700d with D65 source and d/8° analytic geometry in the CIE L*a*b* system, measuring a circular area corresponding to a 6 mm diameter. The parameter L* represents the lightness from 0 (black) to 100 (white); a* denotes the red/green values and b* the yellow/blue values, both ranging from +60 to −60. For each specimen, 25 measures were acquired. Given ΔL*, Δa*, Δb*, the total color difference can be stated as a single value, expressed as
ΔE* = [(ΔL*)^2^ + (Δa*)^2^ + (Δb*)^2^]^1/2^,(1)

The ΔE* value represents the distance of two color points on the CIE L*a*b* color space [41,43].

## Figures and Tables

**Figure 1 gels-07-00111-f001:**
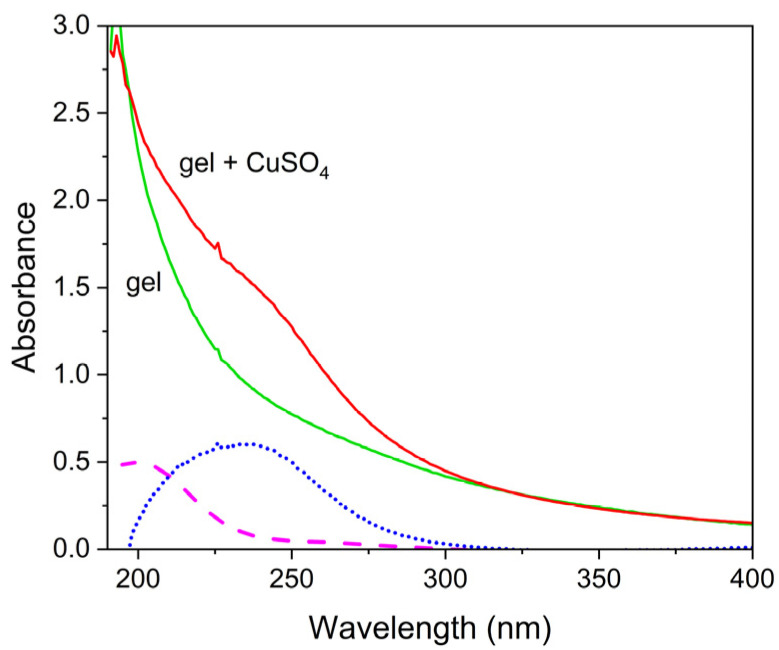
Ultraviolet-visible (UV-Vis) absorbance spectra of pure agar gel and CuSO_4_ in agar gel (full curves), spectrum resulting from the subtraction (dotted curve), and spectrum of CuSO_4_ in free aqueous solution (dashed line). Gel samples are 1 mm thick and the optical path for solutions in the cuvette is 1 cm.

**Figure 2 gels-07-00111-f002:**
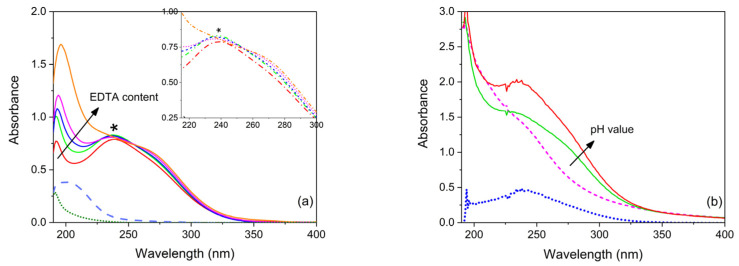
(**a**) UV-Vis absorbance spectra of free aqueous solutions containing CuSO_4_ and ethylenediaminetetraacetic acid (EDTA) at increasing M:L molar ratios (1:1, 1:2, 1:3, 1:4 and 1:10); the spectra of solutions of EDTA (dotted curve) and CuSO_4_ (dashed curve) are also reported for comparison. The asterisk * reports the position of the 239 nm band. (**b**) UV-Vis absorbance spectra of agar gels containing CuSO_4_ and EDTA in a molar ratio of 1:1 at increasing pH (6.0 and 10.0), together with the spectrum resulting from their subtraction (dotted curve). The spectrum of agar gel containing only CuSO_4_ is also reported for comparison (dashed curve). Gel samples are 1 mm thick and the optical path for solutions in the cuvette is 1 cm.

**Figure 3 gels-07-00111-f003:**
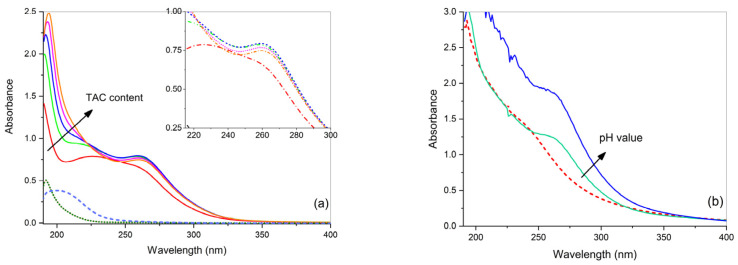
(**a**) UV-Vis absorbance spectra of free aqueous solutions containing CuSO_4_ and ammonium citrate tribasic (TAC) at increasing M:L molar ratios (1:1, 1:2, 1:3, 1:4, and 1:5); the spectra of solutions of TAC (dotted curve) and CuSO_4_ (dashed curve) are also reported for comparison. (**b**) UV-Vis absorbance spectra of agar gels containing CuSO_4_ and TAC in molar ratio 1:2 at increasing pH (5.5 and 10.0); the spectrum of agar gel containing only CuSO_4_ is also reported for comparison (dashed curve). Gel samples are 1 mm thick and the optical path for solutions in the cuvette is 1 cm.

**Figure 4 gels-07-00111-f004:**
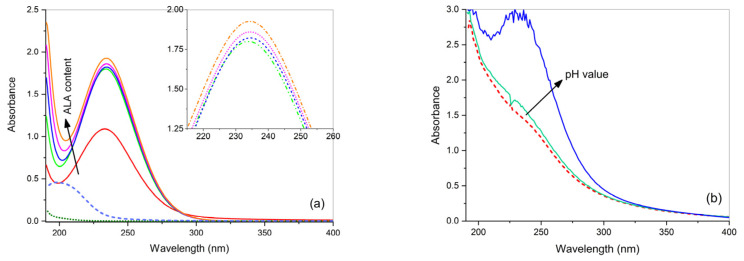
(**a**) UV-Vis absorbance spectra of free aqueous solutions containing CuSO_4_ and alanine (ALA) at increasing M:L molar ratios (1:1, 1:2, 1:3, 1:4, and 1:5) at pH 10.0; the spectra of solutions of ALA (dotted curve) and CuSO_4_ (dashed curve) are also reported for comparison. (**b**) UV-Vis absorbance spectra of agar gels containing CuSO_4_ and ALA in molar ratio 1:2 at increasing pH (3.4 and 10.0); the spectrum of agar gel containing only CuSO_4_ is also reported for comparison (dashed curve). Gel samples are 1 mm thick and the optical path for solutions in the cuvette is 1 cm.

**Figure 5 gels-07-00111-f005:**
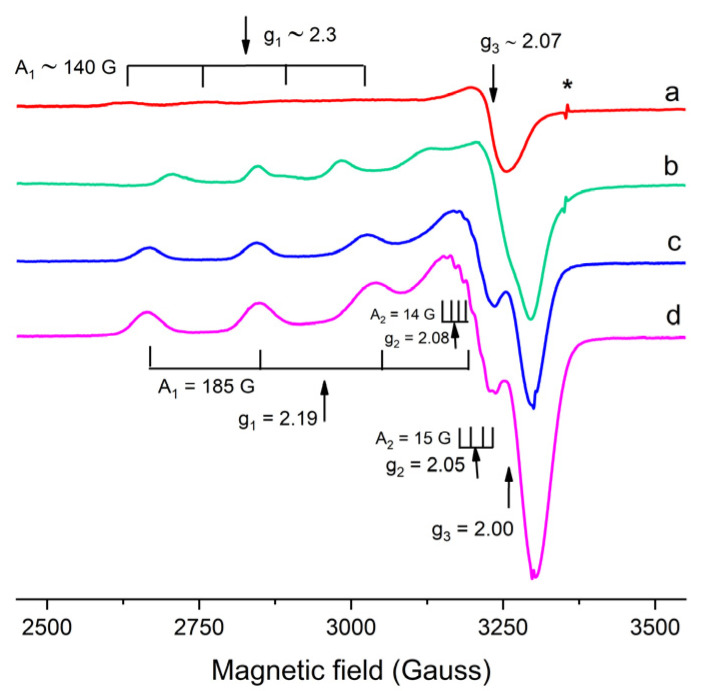
Electron paramagnetic resonance (EPR) spectra of agar gels with CuSO_4_ (**a**) and with additives: EDTA (**b**), ALA (**c**), TAC (**d**), all at pH 10.0. The asterisk * indicates a signal probably due to impurities present in the agar network.

**Figure 6 gels-07-00111-f006:**
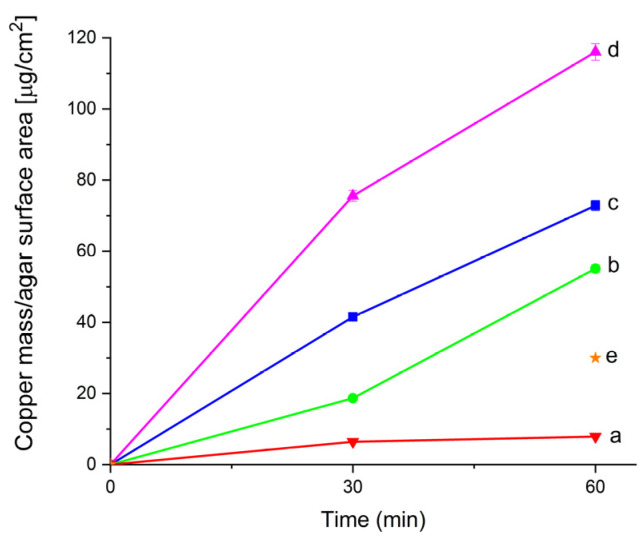
ICP-MS data showing the total copper content per agar surface area of pure agar gel (**a**), with TAC (**b**), with EDTA (**c**), and with ALA (**d**) after 0, 30, and 60 min of contact with laboratory stained specimens at pH 10.0. The asterisk * (**e**) reports the copper content in agar gel with EDTA and with TAC at pH 5.0 and 7.0 respectively.

**Figure 7 gels-07-00111-f007:**
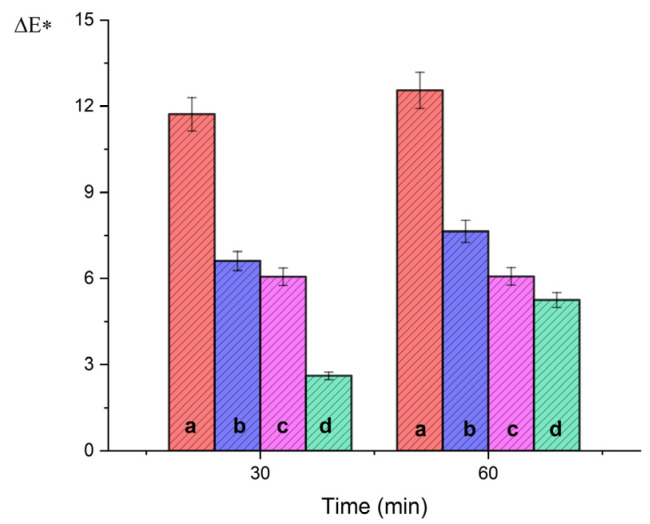
Global color variations ΔE* for stained marble samples after 30 and 60 min contact with gels with ALA (**a**). with TAC (**b**), with EDTA (**c**), and of pure agar (**d**).

**Table 1 gels-07-00111-t001:** pH values and spectral position of the metal-ligand charge transfer (MLCT) bands for precursors and complexes of Cu(II) with ethylenediaminetetraacetic acid (EDTA), ammonium citrate tribasic (TAC), and alanine (ALA), in aqueous solutions, free, and in agar gels, with relative attributions.

Solute	Free			In Agar Gel		
	pH of as Prepared/Modified	λ_max_ (nm)	Attribution	pH of as Prepared/Modified	λ_max_ (nm)	Attribution
CuSO_4_	5.0/n.d.	198	[Cu(H_2_O)_6_]^2+^	5.0/10.0	233	Cu(agar)
CuSO_4_ + EDTA	5.0/6.0, 8.0	239	Cu(EDTA)	6.0/10.0	239	Cu(EDTA)
CuSO_4_ + TAC	9.0/6.0	225 260	Cu(TAC) or Cu_2_(TAC)_2_; Cu(TAC)_2_	5.5/10.0	260	Cu(TAC)_2_
CuSO_4_ + ALA	6.0/10.0	234	Cu(ALA)_2_	3.4/10.0	234	Cu(ALA)_2_

**Table 2 gels-07-00111-t002:** Parameters and attribution of electron paramagnetic resonance (EPR) signals observed at 123 K in agar gels.

Solute	g_1_ ± 0.01	g_2_ ± 0.01	g_3_ ± 0.01	A_1_ (G) ± 5 G	A_2_ (G) ± 5 G	A_3_ (G) ± 5 G	Attribution
CuSO_4_	~2.3	2.07		~140	n.d.		Cu(agar)
CuSO_4_ + EDTA	2.29	2.07		141	n.d.		Cu(EDTA)
CuSO_4_ + ALA	2.18	2.07	2.00	183	12	n.d.	Cu(ALA)_2_
CuSO_4_ + TAC	2.19	2.05	2.00	185	15	n.d.	Cu(TAC) and Cu(TAC)_2_
		2.08			14		

**Table 3 gels-07-00111-t003:** Global color variations (ΔL*, Δa*, Δb*, ΔE*) before (t_1_) and after (t_2_) cleaning of stained laboratory specimens after 30 and 60 min contact.

Cleaning Material	Time (min)	ΔL*	Δa*	Δb*	ΔE*
Pure gel	30	−2.59	−1.18	−0.29	2.61
	60	1.1	5.11	−0.43	5.25
Gel with EDTA	30	3.34	2.81	4.22	6.06
	60	3.56	3.02	3.87	6.07
Gel with TAC	30	4.08	2.92	4.45	6.61
	60	4.92	4.22	4.03	7.64
Gel with ALA	30	6.22	9.9	0.8	11.72
	60	6.22	10.57	2.68	12.55

## Data Availability

Data contained within the article or Appendix A.

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
