# Peer review of "Optimization of Copper Stain Removal from Marble through the Formation of Cu(II) Complexes in Agar Gels"

_gels, 2021, doi:10.3390/gels7030111_

Round 1

Reviewer 1 Report

Interesting article but should be improved on the following points:
Introduction 
L42-43, this sentence " Several cleaning issues are currently open and metal stain removal is one of the unsolved ones." must be changed because it allows to think that the conservators do not know how to do, even if their protocol can be improved, they have already a knowledge and a practice for these problems. Rather than repeating generalities, the authors could specify the current practices to clean marbles, and open on the new perspectives developed recently with the gels. Authors could reduce the number of references cited and keep only the most relevant ones.
L59, the authors cite a study on sol-gel (silica gel). the authors must clarify the differences in chemistry between a sol-gel and an Agar physical gel, as they risk creating confusion and misunderstanding. 
Lacks: the authors wish to improve the understanding of complexes in gels (Cu/chelatant), but they do not provide data to understand the chemistry of these compounds.
Materials and Methods
The preparation and application of the gels refers to other articles of the authors (ref 7 and 1), but it seems essential to mention that the gels underwent a double heating ensuring the formation of the double helix network. I did not understand how they added the solutions (with chelating agent and copper), they add gels in these solution, did they take into account the effects of dilution with the water contained in the gel? It is not clear.
I don't understand how the authors have films at pH 10 when it is known that Agar gels are only stable at neutral pH.
The authors must also justify the choice of their solutions and their protocols, why increase the pH of the solutions of di-sodium EDTA with NH3, whereas the conservators modify the pH of its solutions with the mixtures of di-sodium EDTA and tetrasodium EDTA. The authors must justify their approach.
We do not know the concentrations used for the application of the gels on the marble nor the detail of application of the gel, because the contact between the gel and the marble has an important impact, these data must be specified and not referred to a reference (1) which an act of congres little accessible. 
The authors should give the concentrations of their solutions in molar but also in %massic to be understood by all (chemists prefer molar concentration and conservators use %massic).
Results and discussion
Several times the authors talk about «the aqueous solution containing agar, CuSO4 and chelatant»(like L191, L221), this expression made me doubt the protocols, all these analyses are made on solutions and not on gels, the Agar network is not present? In this case, the authors can't talk about gels. This point must absolutely be solved because it calls into question the whole interest of the article.
L161: Figure 2 does not confirm the explanations given in L161, figures 2, 3 and 4 must be revised: better difference between the curves and add zoom on the studied parts.
The authors present important modifications of the solutions with or without gels: from 5 to 6 for EDTA, from 9 to 5.5 for TAC; but no chemical explanation is given, it seems essential that the authors discuss their results further, adding the behavior of the different compounds as a function of pH:
- EDTA, TAC and ALA stability as a function of pH (example of curve given in attachment)
- CuSO4 stability as a function of pH (Pourbaix diagram)
- Agar gel stability as a function of pH (double helix network or not, available link…)
L151: the authors speak of pH 6 and 8, while on the figure the pH is 6 and 10, there is an error.
In part EPR characterization, the authors speak about the bonds Agar gel with Cu, can they specify the type of bond.
The authors say (L253-254) " However, the Cu(agar) signal is no more observable at pH 10.0, meaning that the ligand outcompetes the agar binding sites at higher pH. " but at this pH the agar gel no longer has its double helix network, the copper would be bound to this network? the authors should discuss their results further. 
Figure 6, to help the reader, the authors should add the neutral pH results presented in reference 7.
The authors say (L297) " the increase in pH does not affect the removal by pure agar gel," I don't agree with this sentence, the authors forget the impact of the pH on the Agar gel network, the impact in practice is that the gel will create more residues, an important problem for conservators
Lacks: the authors should add a section to discuss the impact of pH and the impact of complexation, taking into account the stability of each compound as a function of pH: complexation can more easily take place in a domain where CuSO4 is less stable, by varying the pH, we can also favour the extraction of copper sulphates. 

Reviewer 2 Report

This manuscript could have a significant content but the results need to be presented even nicely with few improvements. I would like to see UV-Vis measurements of Cu:Ligand complexes with a source of brochantite and not copper sulfate which is much more soluble in water.

Round 2

Reviewer 1 Report

the authors have taken into account the comments and made the requested revisions

Author Response

Manuscript ID: gels-1258253

Title: Optimisation of copper stain removal from marble through the formation of Cu(II) complexes in agar gels

Authors: Antonio Sansonetti, Moira Bertasa, Cristina Corti, Laura Rampazzi, Damiano Monticelli, Dominique Scalarone, Adele Sassella, Carmen Canevali*

Gels Horizons: From Science to Smart Materials

Response to Reviewer Comments

Point 1: the authors have taken into account the comments and made the requested revisions

Response 1: We thank the Reviewers for all their comments, which helped on paper improvement.